# Molecular Basis of Accelerated Aging with Immune Dysfunction-Mediated Inflammation (Inflamm-Aging) in Patients with Systemic Sclerosis

**DOI:** 10.3390/cells10123402

**Published:** 2021-12-02

**Authors:** Chieh-Yu Shen, Cheng-Hsun Lu, Cheng-Han Wu, Ko-Jen Li, Yu-Min Kuo, Song-Chou Hsieh, Chia-Li Yu

**Affiliations:** 1Department of Internal Medicine, National Taiwan University Hospital, Taipei 10002, Taiwan; tsichhl@gmail.com (C.-Y.S.); b89401085@ntu.edu.tw (C.-H.L.); chenghanwu@ntu.edu.tw (C.-H.W.); dtmed170@gmail.com (K.-J.L.); 543goole@gmail.com (Y.-M.K.); 2Graduate Institute of Clinical Medicine, National Taiwan University College of Medicine, Taipei 10002, Taiwan

**Keywords:** systemic sclerosis, cellular senescence, inflamm-aging, chromosome instability, oxidative stress, proinflammatory cytokines, anti-centromere antibody, anti-topoisomerase 1 antibody, immune functional adaptation, tissue fibrosis

## Abstract

Systemic sclerosis (SSc) is a chronic connective tissue disorder characterized by immune dysregulation, chronic inflammation, vascular endothelial cell dysfunction, and progressive tissue fibrosis of the skin and internal organs. Moreover, increased cancer incidence and accelerated aging are also found. The increased cancer incidence is believed to be a result of chromosome instability. Accelerated cellular senescence has been confirmed by the shortening of telomere length due to increased DNA breakage, abnormal DNA repair response, and telomerase deficiency mediated by enhanced oxidative/nitrative stresses. The immune dysfunctions of SSc patients are manifested by excessive production of proinflammatory cytokines IL-1, IL-6, IL-17, IFN-α, and TNF-α, which can elicit potent tissue inflammation followed by tissue fibrosis. Furthermore, a number of autoantibodies including anti-topoisomerase 1 (anti-TOPO-1), anti-centromere (ACA or anti-CENP-B), anti-RNA polymerase enzyme (anti-RNAP III), anti-ribonuclear proteins (anti-U1, U2, and U11/U12 RNP), anti-nucleolar antigens (anti-Th/T0, anti-NOR90, anti-Ku, anti-RuvBL1/2, and anti-PM/Scl), and anti-telomere-associated proteins were also found. Based on these data, inflamm-aging caused by immune dysfunction-mediated inflammation exists in patients with SSc. Hence, increased cellular senescence is elicited by the interactions among excessive oxidative stress, pro-inflammatory cytokines, and autoantibodies. In the present review, we will discuss in detail the molecular basis of chromosome instability, increased oxidative stress, and functional adaptation by deranged immunome, which are related to inflamm-aging in patients with SSc.

## 1. Introduction

Systemic sclerosis (SSc) is a complex multi-system autoimmune disease characterized by chronic inflammation and tissue fibrosis in the skin and internal organs (especially the lung) [1,2,3]. Despite extensive investigations, the underlying molecular mechanism of SSc remains unclear until now. In general, this heterogenous autoimmune disorder is featured by immune dysregulation, inflammation, and endothelial cell dysfunction followed by defective vascular repair, neovascularization, and finally tissue fibrosis [4,5]. In clinical practice, the endothelial dysfunction manifested as Raynaud’s phenomenon is the first clinical presentation in the patients due to damage in the finger capillary lumen [6,7]. Furthermore, vascular injury may induce fibroblast activation and tissue fibrosis, resulting in irreversible scarring and organ failure [8,9,10]. Similarly to the other systemic autoimmune diseases, the pathogenesis of SSc includes genetics/epigenetics, transcriptomics/proteomics/metabolomics, environmental/stochastic factors, immune dysfunction-mediated inflammation, and the factors mediating cell senescence or inflamm-aging [4,5,10,11,12]. The latter factors include excessive oxidative stresses [13,14,15,16], chronic inflammation [17,18], vascular insufficiency [19,20], autoantibody formation [21], DNA damage/DNA damage responses [22], and telomere shortening [23]. In addition, the cancer incidence in these patients also increased [24,25,26] in association with inflamm-aging [27,28]. It is conceivable that inflammatory cytokines (IL-1β, IL-6, and TNF-α) can enhance oxidative stresses in causing endothelial cell dysfunction and mitochondrial functional impairment. This early vascular aging may further accelerate inflamm-aging in patients with SSC. In the present review, we first discuss general pathogenesis and then the detailed immunopathological/immunobiological bases of inflamm-aging in the patients with SSc.

## 2. The Roles of Genetics, Environmental Risk Factors, Stochastic Processes, and Epigenetics in the Pathogenesis of Patients with Systemic Sclerosis

It is generally agreed that patients with SSc exhibit the phenotypes of SSc, which is induced in genetically susceptible individuals after exposure to the environmental factors such as silica or viruses [29,30,31]. Although genetic factors, especially the MHC class II, may play important roles in the development of SSc, the concordant rate of SSc in monozygotic twins is only 4.27% [32]. This rate is much lower than rheumatoid arthritis (12.3%) [33], multiple sclerosis (16.7%) [34], systemic lupus erythematosus (25%) [35], or primary biliary cirrhosis (77%) [36]. This evidence may suggest the importance of gene-specific epigenetic alterations in the pathophysiology of these patients. Thus, a potential role of environmental-epigenetic hypothesis in the pathogenesis of SSc was advocated [37]. We discuss these upstream factors in the following sections.

### 2.1. Genetic Predisposition in Patients with SSc

Similarly to the most autoimmune diseases, SSc is not elicited by a single gene polymorphism but is elicited by the cumulated results of the polygenic variants that predispose the individual to the development of overt disease. This may indicate that different HLA and non-HLA genetic foci are involved in susceptibility to overt SSc.

#### 2.1.1. The Role of Major Histocompatibility Complex (MHC) Class II

MHC class II represents one of the most significant genetic loci in the development of SSc, particularly *HLA-DRB1, DPB1, DQA1, DQB1*, and *HPB1* [38,39,40]. The importance of these MHC-II loci on haplotypes has been found correlation with clinical subsets of patients with SSc in different races. In clinical practice, lung fibrosis was commonly found in patients with SSc who carry the *HLA-DPB1*03:01* haplotype. In contrast, SSc in which the *HLA-DPB1*4* haplotype is carried was found in ACA-positive patients rather than ACA-negative patients [41]. Evidence also demonstrates that the *HLA-DQ5-HLA-DR1* haplotype is associated with low progression rates from early SSc to established SSc [42]. In addition, *HLA-DQA1*5* [43] and *HLA-DRB1*10* [44] are identified as risk factors for juvenile-onset SSc (JSSc) [43] other than adult-onset SSc. The results from immunochip data analysis confirmed a strong association between the HLA region and SSc in that seven SNPs and polymorphic amino acid positions in HLA region were associated with either SSc or specific autoantibody profiles [40].

#### 2.1.2. The Role of Non-HLA Genes

Lopez-Isac et al. [45] conducted a GWAS and meta-analysis that allowed identification of 13 new risk loci for SSc. Many of these different SSc-associated genes found in non-HLA regions are found to possess strong association with genes related to the immune system. Jin et al. [46] demonstrated that a total of 47 genes or specific genetic regions were related with SSc. These genes include *HLA, STAT4, CD247, TBX21, PTPN22, TNFSF4, IL23R, IL2RA, IL-21, SCHIP1/IL12A, CD226, BANK1, C8orf13-BLK, PLD4, TLR-2, NLRP1, ATG5, IRF5, IRF8, TNFAIP3, IRAK1, NFKB1, TNIP1, FAS, MIF, HGF, OPN, IL-6, CXCL8, CCR6, CTGF, ITGAM, CAV1, MECP2, SOX5, JAZF1, DNASEILE3, XRCC1, XRCC4, PXK, CSK, GRB10, NOTCH4, RHOB, KIAA0319, PSD3*, and *PSOR1C1.* These data support that SSc is a complex disease associated mainly with immune regulatory and inflammatory genes. Moreover, Ota et al. [47] found that the associated genetic variants are mainly located in non-coding regions that can affect the gene expression, including those associated with innate immunity, adaptive immune response, or cell death. In contrast, there are only a few genes involved in the fibrotic process or vascular homeostasis. Beretta et al. [48] explored genome-wide whole blood transcriptome profiling in European SSc patients and observed in that 15224 genes and 1277 functional pathways were available. The authors found that these gene expressions were connected with a dysregulation in type-I interferon, Toll-like receptor cascade, tumor suppressor p53 protein function, and the platelet degranulation/activation. By using bioinformatic approaches, Sun et al. [49] identified the key hub genes and microRNAs (miRs) for modulating the occurrence and development of SSc. In total, 783 expressed genes were identified of which 770 genes (142 upregulated and 628 downregulated) were matched to the genes in SSc skin samples. Further gene ontology analysis indicated that the upregulated genes were mainly involved in immune responses, and the downregulated genes were rich in glycinergic synaptic transmission. In the protein–protein interaction network, 22 nodes were selected as key genes, including several members of chemokine families. It is quite interesting that hsa-miR-26b-5p can target CXCL9 and CXCL13. Moreover, the hsa-miR-26b-50 inhibitor inhibits fibrosis in TGF-β-activated fibroblasts. Recently, Jafarinejad-Farsangi et al. [50] discovered that miR-21 and miR-29a could modulate the expression of collagen in dermal fibroblasts of patients with SSc.

### 2.2. Environmental Risk Factors and Stochastic Processes in Patients with SSc

In addition to genetic predisposition, the environmental etiology of SSc has also been extensively investigated since clusters of the disease have been identified among certain occupational groups such as gold miners [51] and silica-induced silicosis [52]. A variety of environmental agents including vinyl chloride [53], hydrocarbons [54], epoxy resins [55], rapeseed oil [56], medications [57], and even heavy metals [58] have been reported. A comprehensive review of potential occupational and environmental factors in SSc onset has been reported by Marie et al. [58]. It is believed that scleroderma is elicited by both genetic and environmental/occupational factors. However, random events or stochastic processes also play pivotal roles in SSc pathogenesis similar to other systemic autoimmune diseases. Roberts-Thomson P.J. and Walker J.G. [59] speculated that these stochastic processes might result in acquired somatic mutations or epigenetic alterations that affect the genes coding for immune receptors, tolerogenic threshold, or proteins involved in immune regulation, inflammation, and/or repair. Genetic predisposition, genetic variants in non-coding regions for regulating gene expression, and environmental/stochastic factors related to SSc are summarized in Figure 1.

### 2.3. Aberrant Epigenetic Expression in Patients with SSc

Epigenetics includes DNA methylation, histone modifications, and non-coding RNA transcripts [60]. DNA methylation is a biochemical process for the transfer of a methyl group onto the C5 position of cytosine with S-adenosyl-methionine (SAM) as the methyl donor to form 5-methylcysosine (5mC) at the position of a repeated CpG dinucleotides (CpG island) in the promoter region of a gene for repressing its transcription [61]. In contrast, the demethylation of DNA activates genes transcription [62]. The methylation of DNA is mediated by DNA methyltransferase (DNMT) 1, 3a, and 3b [63]. Conversely, gene transcription is achieved only after DNA demethylation by ten-eleven transformation enzymes (TET), TET1, TET2, and TET3 to become 5-hydroxymethylation (5hmC) [63]. Another pathway of DNA demethylation was achieved by APOBEC-mediated deamination of 5mC [64]. On the other hand, the post-transcriptional modifications of amino acid residues in histone may change the chromosome structure and become another epigenetic mechanism of gene expressions. The histone modifications include acetylation, methylation, phosphorylation, deamination, ADP ribosylation, ubiquitination, sumoylation, and β-N acetyl glucosamination. These modifications can change the charge of histone and affect chromatin structure and gene expression [65]. The most common modification of histone is acetylation catalyzed by histone acetyltransferases (HATs) and histone deacetylases (HDACs) in that HATs transfer an acetyl group from acetyl-CoA to the ε–amino group of lysine side chains. On the contrary, HDAC removes an acetyl group from the lysine tail. The acetylated state of histone neutralizes its positive charge that weakens the interaction between histone and DNA strand and vice versa [66].

#### 2.3.1. Abnormal DNA Methylation and Histone Modifications in the Immune-Related Cells of SSc Patients

Lei et al. [67] found that global hypomethylation in CD4+T cells in SSc patients was a result of a decrease in DNMT1 and methyl-CpG-binding domain proteins (MBD): MBD3 and MBD4. More specifically, Wang et al. [68] and Almanzar et al. [69] demonstrated that DNA hypermethylation in the FOXP_3_ promoter subsequently impaired the functions of regulatory T cells (Treg) and caused hyperactivity of CCR6+ Th cells. Coit et al. [70] identified a total of 105 and 144 differentially methylated sites and genes unique for either juvenile SSc (JSSc) and localized SSc including FGFR2, STAT3, NF-κB, IL-15, and NOTCH3 pathways. Baker Frost et al. [71] intended to investigate the association of DNA methylation levels in dermal fibroblasts obtained from patients of African ancestry and found widespread reduced DNA methylation of DLX5 and TMEM140. Rezaei et al. [72] suggested that hypomethylation of IRF7 promotor might play a role in SSc pathogenesis via promoting the IRF7 expression in PBMCs of patients with SSc. Furthermore, Ding et al. [73] unveiled that hypomethylation and the subsequent upregulation of type I interferon-associated genes play a critical role in SSc pathogenesis. Chen et al. [74] concluded that hypomethylation of IFN-related genes in CD4^+^ T cell is a common epigenetic feature of autoimmune disease including Grave’s disease, rheumatoid arthritis, systemic lupus erythematosus, and SSc patients, and the DNA methylation profile of IFN-related genes could become promising biomarkers for autoimmune diseases.

Wang et al. [75] demonstrated that the global histone H4 hyperacetylation and global histone hypomethylation in SSc B cells could correlate with skin thickness and increased disease activity. Kramer et al. [76], by using 3-deazaneplanocin A (DZNep) to inhibit H3lys27 histone methylation, found H3K27 histone de-trimethylation activated fibroblasts and induced fibrosis by upregulating fra-2 expression. Ciechomska et al. [2,77] further demonstrated that histone demethylation together with TLR-8 activation in monocytes could promote trans-differentiation of fibroblasts to myofibroblasts (MFB) via an activator protein 1 family member, fra-2. van der Kroef et al. [78] explored aberrant H3K4me3 and H3K27ac marks in SSc monocytes and found that they correlated with their IFN signatures. Enzymes modulating these reversible marks could restore monocyte homeostasis in SSc. Aberrant DNA methylation and histone modifications in mediating SSc pathogenesis are illustrated in Figure 2.

#### 2.3.2. Aberrant Non-coding RNA (ncRNA) Expression in Patients with SSc

Aberrant ncRNA expressions in SSc including microRNAs (miRs) and long non-coding RNAs (lncRNAs) have been extensively reviewed in the literatures [10,31,79,80,81]. Tsai et al. [10] reviewed the literatures and found an intriguing imbalance relationship between pro-fibrotic miRs and anti-fibrotic miRs towards profibrotic status in SSc. The pro-fibrotic miRs include miR-21, miR-29, miR-129-5P, miR-133, miR-142-3p, miR-503, and let-7a. The anti-fibrotic miRs include miR-15b, miR-16, miR-27a, miR-27B, miR-29a, miR-132, miR-145, miR-146, miR-150, miR-196a, and miR-335. In addition, the differential expression levels of lncRNAs included TSIX, NRIR, ncRNA00201, OTUD6B-AS, CTBP1-AS2, AGAP2-AS1, and HIFA-AS1, which can target mRNAs and might also involve pathogenesis and pathological changes in patients with SSc.

The aberrant non-coding RNA expression and relevant immunopathological changes in patients with SSc are depicted in Figure 3.

## 3. The Molecular Basis of Inflamm-aging (Immune Senescence) in the Elderly People

Inflamm-aging is an age-associated inflammatory status that is regarded as one of the most striking consequence of immunosenescence. Salvioli et al. [82] defined inflamm-aging as the chronic subclinical increased production of proinflammatory mediators such as IL-1β, IL-6, and TNF-α by immune cells of elderly persons independent from immunological stimuli. Synonymously, immune-senescence is a striking feature of immune system aging by increased percentage of late effector and memory T cells, marked shrinkage of T cell repertoire, decreased numbers and frequencies of B cells, and increased myeloid cells as well as natural killer (NK) cells [83,84]. However, by considering inflamm-aging and human longevity in the “omics” era, Monti et al. [85] concluded that (1) inflamm-aging contains a structure in which specific combinations of pro-inflammatory and anti-inflammatory mediator are involved; (2) many organs, tissues, and cell types other than immune cells participate in producing pro-inflammatory and anti-inflammatory stimuli; and (3) inflamm-aging is a dynamic that can be propagated locally to neighboring cells and systemically from organ to organ by circulating products and microvesicles. In considering the immune system as an important part of the larger neuro-endocrine-immune axis, Fulop et al. [86] summarized that inflamm-aging in aging immune systems aims to maintain and compensate the general immune homeostatic functions by appropriately improving immune-inflammatory function. Accordingly, immunosenescence and inflamm-aging may act as two sides of the same coin.

Chronic stresses are the major contributors to accelerated aging in an immune system. These chronic stresses include low-grade chronic proinflammatory states [87,88] and oxidative stresses [86,89] that are linked to impairing telomerase function and subsequent telomere shortening [90]. It is conceivable that the chronic stresses would stimulate non-coding RNA expression to link inflamm-aging, cellular senescence, and cancer. Although chronic inflammation contributes to the continuous activation of immune-related cells, the cell senescence process involves the acquisition of a senescence-associated secretory phenotype. It is also believed that some signaling pathways such as NF-κB, mTOR, sirtuins, TGF-β, and Wnt are related to inflammation, cellular senescence, cancer, and age-related disease [91,92]. Olivieri et al. [91] further classified some miRs to senescence-associated miRs (SA-miRs), inflammation-associated miRs (inflamm-miRs), and cancer-associated miRs (onco-miRs). Teodori et al. [93] in their in vitro study observed that inflamm-miRs could modulate metabolic pathways such as lysine degradation and lengthening of fatty acids, which are linked to the modulation of microbiota composition, including *prevotella, ruminococcus*, and *oscillibarter* species.

In addition to intracellular inflamm-miRs, small extracellular vesicles contain miRNA cargo with anti-apoptotic members of SA-secretory phenotype exerting immunomodulatory effects on inflamm-aging for maintaining a health aging state [94,95]. Rippo groups [96,97] proposed that nuclear-encoded SA-RNAs can translocate to mitochondria (SA-mitomiRs), including let-7b, miR-146A, -133B, -106a, -19b, -20a, -34a, -181a, and -221, in affecting the energetic oxidative and inflammatory functions of senescent cells. These modulations can further enhance the loss of mitochondrial integrity and functions of aging cells toward inflammatory response.

## 4. The Molecular Basis of Accelerated Inflamm-aging in Patients with SSc

Senescence cells can secrete a number of proteins collectively called the senescence-associated secretory phenotype (SASP) [98,99]. SASP contains growth factors, proinflammatory cytokines, and extracellular proteases acting as a double-edged sword to mediate the following: (1) the recruitment of immune system to the premalignant lesions for repairing the damaged tissues and (2) the secretion of many proinflammatory factors such as IL-6, IL-8, macrophage inflammatory proteins (MIPs), and membrane cofactor proteins (MCPs) to elicit inflammation [100]. The signaling pathway underlying the SASP includes DNA damage response (DDR) [101], stress kinase, inflammasome, alarmins, cell survival-related transcription factors, miRNA, RNA stability, autophagy, and metabolic regulators [102]. Alamins represent another emerging potential family of inflammation-associated early phase SASP regulators. These molecules are easily released from the chromatin and actively secreted at senescence or during cell necrosis. The chromatin-associated high mobility group box 1 protein (HMGB1) is the first found alamin acting as both an early phase SASP regulator and SASP factor [101]. A diagrammatic sketch of inflamm-aging related to chronic stresses and its immunopathological processes is shown in Figure 4.

The patients with SSc can fulfill the above-mentioned pathological criteria of SASP and thereby the definition of inflamm-aging. We will discuss in detail the molecular bases of accelerated inflamm-aging.

### 4.1. Role of Increased Chromosome Instability, DNA Damage, and Telomere Shortening in Accerelted Infalamm-aging in SSc

Emerit et al. [22] observed a diversity of chromosome abnormalities including gaps/breaks of one or both chromatids, acentric or dicentric fragments, ring chromosomes and other pattern in lymphocytes, fibroblasts, or bone marrow preparations from SSc. Wolff et al. [103] observed that SSc lymphocytes exhibited a generalized susceptibility to DNA damage by free radicals either spontaneously or medication-induced. This observation was further supported by Takeuchi et al. [104]. Roberts-Thomson et al. [105] found that SSc exhibited a higher proportion of mitotic recombination mutations supporting the acquired genetic damage in SSc patients. Artlett et al. [106] demonstrated that the average loss was more than 3 kb length telomeric DNA in SSc patients and their family members than the controls when exposed to common environmental factors such as air pollutants, medications, pesticides, or fungicides. However, a conflicting result of telomere length increase in limited scleroderma patients was also reported by MacIntyre et al. [107]. Lakota et al. [108] revealed that short telomere was only found in lymphocytes, but not in granulocytes, in association with interstitial lung disease (ILD). This acquired cell lineage-specific telomere shortening may be a consequence of lung disease rather than its pathogenic driver. The genomic instability in SSc patients may explain a two-fold higher incidence of cancer than in the general population [109].

In order to explore the cause of telomere erosion in SSc, Ohtsuka et al. [110] unveiled the polymorphism of telomerase RNA component gene in that the frequency of the A/A alleles in SSc (18.9%) was significantly higher than normal controls (18.9% vs. 5.1%) compared with G/G (35.8% vs. 40.8%), G/A (45.37% vs. 54.1%), and G/G+G/A (81.7% vs. 94.9%) in SSc skin-derived fibroblasts. Accordingly, telomerase activity markedly decreased in SSc reported by Tarhan et al. [111]. Moreover, Martelli Palomino et al. [112] evaluated two DNA repair genes, XRCC1Arg399Gln and XRCC4Ile401Thr, in SSc patients. The authors noted that the XRCC1Arg399Gln allele only increased in healthy controls, whereas the XRCC4Ile401Thr allele became enhanced in both SSc and control groups with DNA damage. Furthermore, the XRCC1Arg399Gln allele was also associated with the presence of anti-nuclear antibodies and anti-centromere antibodies but was not associated with clinical features. Vlachogiannis et al. [113] discovered that increased DNA damages in SSc patients under cytotoxic treatment were displayed as increases in oxidative stress and abasic sites, defective DSB/R (denoted by downregulation of MRE11A and PRKDC) and base excision repair (denoted by PARP1 and XRCC1), and the upregulation of apoptosis-related genes (BAX and BBC3). In addition, the damage in SSc PBMC also significantly correlated with the expression of type I interferon-induced genes (denoted by IFIT1, IFI44, and MX1). In short conclusion, these results may suggest that defective DDR accelerates type I interferon expression, contributing to tissue inflammation and fibrosis in SSc patients.

### 4.2. Roles of Increased Oxidative Stress in Accelerating Inflamm-aging of Patients with SSc

Increased generation of intracellular reactive oxygen species (ROS) was found in patients with SSc [114,115,116,117,118]. However, Ogawa et al. [119] demonstrated that around 25% SSc patients had increased anti-oxidants levels in the serum. Savas et al. [114] found that one-quarter of the patients showed high serum oxidants, but no difference in the total anti-oxidant capacity was observed. The oxidative stress-related molecules in the serum included urinary 8-oxodG levels [116] and isoprostanes [120,121], N^ε^-hexanoyl-lysine [122], and heat-shock protein 70 [123] in the serum. These oxidative molecules not only induced oxidative stress but enhanced collagen synthesis by human pulmonary smooth muscle cells [124]. Recently, Giordo et al. [125] found that Iloprost (a stable analog of prostacyclin) could attenuate SSc serum-induced collagen synthesis by human pulmonary microvascular-endothelial cells via anti-oxidant mechanisms.

The NADPH oxidase (NOX) family of the membrane-associated enzymes can catalyze the reduction of O_2_ molecules to form inducible ROS. Currently, seven NOX isoforms (NOX1-NOX7) have been identified in humans, with substantial differences in their tissue distribution [126]. Piera-Velazquez et al. [127] disclosed increased NOX_4_ expression in SSc-dermal fibroblasts regulated by transforming growth factor β (TGF-β). Spadoni et al. [128] found that both NOX2 and NOX4 in SSc fibroblasts can generate ROS, and they were maintained by a ROS-mediated loop to induce cell activation and DNA damage. Amico et al. [129] explored and observed that high levels of ROS in SSc-T lymphocytes are catalyzed by NOX via ERK1/2 phosphorylation.

NOX-mediated oxidative stress can induce cellular senescence in SSc patients via inflammation, autoimmunity, and tissue fibrosis [12,130,131]. On the other side, the pathological levels of mitochondrial ROS (mtROS) can also promote mtDNA damage and decreased expression of protein related to DNA repair including 8-oxoguanine DNA glocosylase 1 (OGG1) and sirtuin 3 (SIRT3, a histone deacetylase) [132]. Movassaghi et al. [133] further revealed that the mean mtDNA copy number was lower in SSc patients due to oxidative damage. Wyman et al. [134] even found that declining SIRT levels and activity are related to accelerated aging in SSc patients. The roles of chromosome instability and oxidative stress in inflamm-aging of patient with SSc are illustrated in Figure 5.

### 4.3. Immunopathological/Inflammatory Basis of Accelerated Inflamm-aging in Patient with SSc

Inflamm-aging denotes the state of upregulation of the inflammatory response at older ages mediated by increased circulating proinflammatory cytokines including IL-1, IL-6, TNF-α, IL-12, IFN-γ, and IFN-β [135,136,137]. Recently, IL-17 was found to be further implicated in the inflamm-aging formation [138,139]. Immunopathological/inflammatory mechanisms in accelerating inflamm-aging of SSc are discussed in detail below.

#### 4.3.1. Aberrant Pro-inflammatory Cytokine Expression in Mediating Inflamm-Aging in SSc

Elevated serum TNF-α level in patients with SSc was associated with pulmonary fibrosis, decreased vital capacity, and pulmonary arterial hypertension [140,141]. The TNF-α inhibitor can improve endothelial functions and decrease the risk of pulmonary arterial hypertension in SSc [142]. Janardan et al. [143] found that the first intron of TNF-β gene, TNF-β^+252^ locus played an important role in the etiopathogenesis of SSc. Moreover, Lomeli-Nieto et al. [144] demonstrated the polymorphic TNF-α promoter regions, TNF-α^−308^ and TNF-α^−258^, in SSc patients in that TNF-α^−308^ G > A and TNF-α ^−238^ G > A polymorphisms were associated with higher serum levels of TNF-α and anti-RNA polymerase III antibody. In addition to TNF-α, Hasegawa et al. [145] and Gourh et al. [146] demonstrated that the elevated serum levels of Th2 cytokines IL-4, IL-10, and IL-13 are also associated with autoantibody production and clinical manifestations in patients with SSc.

The IL-1 family cytokines comprise seven members with proinflammatory activity (IL-1α, IL-1β, IL-18, IL-33, IL-36α, IL-36β, and IL-36γ) and four members with anti-inflammatory activity (IL-1Ra, IL-36Ra, IL-37, and IL-38) [147]. These IL-1 family cytokines play a crucial role in SSc pathogenesis, particularly in myocardial inflammation, dysfunction, and fibrosis [148,149,150].

Hasegawa et al. [151] firstly discovered that the serum levels of IL-6 were significantly elevated in early diffuse type SSc patients with pulmonary fibrosis. In contrast, serum sIL-6R levels were found higher in patient with limited cutaneous SSc. The high serum levels of IL-6 reflect elevated C-reactive protein (CRP) levels in SSc patients [151,152]. Denton et al. [153] confirmed that IL-6 blockades could reverse TGF-β pathway activation in dermal fibroblasts to suppress skin fibrosis. However, Shima Y. et al. [154] unveiled that SSc patients with high IL-13 serum levels abrogated the influence of IL-6 on tissue fibrosis. Recently, Iorio et al. [155] suggested that anti-IL-6 therapy should be considered against inflamm-aging in the elderly.

IL-17 comprises a family of cytokines IL-17A to IL-17F in that IL-17A is the most abundant one against microbial infections and is involved in inflammation and autoimmunity. Brembilla et al. [156] documented increased IL-17A in serum and the target organs of SSc patients. Lei et al. [157] found that Th_17_ cells and secreted IL-17 participated in skin and lung fibrosis in a bleomycin-induced SSc murine model via enhancing fibroblast proliferation. Robak et al. [158] further discovered that increased serum levels of IL-17B, IL-17E, and IL-17F, but not IL-17A, play a role in SSc pathogenesis. In contrast, IL-17F is associated with limited form SSc. Moreover, many molecules including programmed death protein-1 (PD-1), phosphatase SHP2 and STAT3, intracellular signaling molecules Ras/Erk, mTOR, and complement components are all involved in SSc-associated fibrosis [159]. In clinical practice, a reversible S-adenosyl-l-homocysteine hydrolase (SAHH) inhibitor, DZ2002, can prevent the development of experimental dermal fibrosis by suppressing a number of cytokines and growth factors involved in dermal fibrosis [160]. Lu et al. [161] demonstrated that thalidomide could effectively prevent skin and pulmonary fibrosis in a mouse model via inhibiting the TGF-β/Smad 3 signaling pathway and rectifying predilection of Th_17_/Treg balance in SSc.

Inflamm-aging is also found in other chronic inflammatory diseases including periodontitis, osteoporosis, and arthritis [162,163,164]. Interferon–γ (IFN-γ) plays a key role in inflammation-related bone loss [165,166]. Tan et al. [167] investigated the differential expression of 18 interferon-inducible genes and vasculotropism in the peripheral blood cells of SSc patients and found that six of them were identical to those of SLE patients. Wu et al. [168] confirmed that upregulation of interferon regulatory factor 7 (IRF7) in SSc skin can interact with Smad3 and potentiates TGF-β-mediated fibrosis. This result implicates that type I interferon signaling is involved in SSc pathogenesis. Recently, Tan et al. [169] found that inflamm-aging-related cytokines, IL-17 and IFN-γ, may accelerate periodontal destruction.

#### 4.3.2. Autoantibody-mediated Inflamm-aging in SSc Patients

The circulating specific autoantibodies are detected in 90–95% of SSc patients and become the useful diagnostic biomarkers in clinical practice. These specific autoantibodies, including anti-TOPO-I, ACA, anti-RNPIII, anti-ribonuclear proteins (anti-U1, anti-U3, and anti-U11/U12 RNP), and anti-nucleolar antigen (anti-Th/T0, anti-NOR90, anti-Ku, anti-RuvBL1/2, and anti-PM/Scl), link to distinct clinical manifestations [10,21,170]. Shen et al. [171] found that ACA and anti-TOPO-I-containing sera from SSc patients with Raynaud’s phenomenon could induce vascular endothelial cell senescence via decreasing relative telomerase content and increasing β-galactosidase but not via the classical p53-p21 senescence pathway. Raschi et al. [172] further confirmed that the purified scleroderma-specific autoantibodies containing immune complexes could engage Toll-like receptors via their nucleic components to elicit inflammation, endothelial damage, and pro-fibrotic cascades.

Recently, Zhao et al. [173] identified a novel anti-tubulin-α-1c autoantibody in SSc that was correlated with inflammation, Raynaud’s phenomenon, and the titer of anti-Scl-70, anti-CENP, and anti-cardiolipin autoantibodies in SSc patients. Furthermore, Adler et al. [174] discovered another novel autoantibody that can target telomere-associated proteins including telomere-specific reverse transcriptase (hTERT) and telomerase RNA component (hTR). These findings can explain a subset of SSc patient expressing shortened peripheral leukocyte telomere length and fibrotic lung disease mimicking inflamm-aging.

## 5. Abnormal Immunobiological Basis of Accelerated Inflamm-aging in SSc Patients

A number of immune cell types were found to have implications in SSc pathogenesis including T cells, B cells, dendritic cells, mast cells, and macrophages. The interactions among different immune cell types in mediating the inflamm-aging of SSc are intriguing. We will discuss in detail the abnormal immunobiological bases of B and T lymphocyte subsets relevant to inflamm-aging.

### 5.1. The Role of Abnormal B Lymphocyte Homeostasis and Functions in the Inflamm-aging of SSc Patients

A recent study by Saito et al. [175] demonstrated that downregulation of B cell function improved skin fibrosis in a genetic model of SSc tight skin mouse. The same group determined the phenotypes and functional abnormalities of blood B cell subsets in patients with SSc [176]. They found that SSc patients expressed expanded naïve and activated B but had diminished memory B cells. Moreover, the CD19-overexpressed memory B cells are related to their hyper-reactivity. For further elucidating the cause of CD19 molecule overexpression, the same group assessed the GT repeat polymorphism in the 3′-untranslated region (3′-UTR) of CD19 genes [177]. They unveiled that these alleles were in linkage disequilibrium. The -499T allele played a primary role in that CD19-499 G > T polymorphism is associated with higher CD19 expression on both naïve and memory B cells and with susceptibility to SSc. Simon et al. [178] further analyzed memory B cells subsets into CD19^+^IgD^-^CD27^-^CD38^+^ double negative (DN)-1, CD19^+^IgD^lo^CD27^+^CD38^+^ unswitched, CD19^+^IgD^-^CD27^+^CD38^+^CD95^-^ resting switched, and CD19^+^IgD^-^CD27^-^CD38^-^CD95^+^ active switched memory (ASM) B cells (ASM-B) in SSc patients. These authors noted that the increase in ASM-B cells is associated with a severe form of SSc patients with pulmonary fibrosis. Functionally, the correlation of ASM B cells acting as effector memory-plasma cell precursors with anti-TOPO-I antibody production can reflect their pathological roles in SSc pathogenesis. The pathological role of B lymphocytes in SSc has been also critically reviewed in the literature [8,179].

### 5.2. The Roles of Abnormal T Lymphocyte Subsets in the Accelerated Inflamm-aging of SSc Patients

T cells activation plays a key role in the development of vascular damage and tissue fibrosis in SSc patients. Th1 or Th2 predilection contributes to the activation of proinflammatory (Th1) or pro-fibrotic (Th2) response [180]. However, there is no agreement on T cell subpopulation count in SSc patients. Many controversies exist regarding the CD4^+^ T cell population in SSc. Fiocco et al. [181] found that CD4^+^CD26^+^ and CD4^+^CD25^+^ percentage and absolute number increased and CD8^+^CD29^+^ percentage in SSc patients decreased, correlating with disease activity. For exploring the mechanism of elevated serum IL-13 levels in SSc patients, Medsger et al. [182,183] found that IL-13 was not only excessively produced by peripheral CD8^+^T cells but GATA-3 upregulation in these CD8+ T cells can become a biomarker of immune dysfunction in SSc patients. Frantz et al. [184] concluded a general agreement in the decreased functional ability of circulating regulatory T cells (Treg) in SSc. Gumkowska-Sroka et al. [185] observed higher total lymphocyte count with increased suppressor T cells (Ts), helper T cells (Th), and double positive T (Tdp) but also observed a reduction in double negative T (Tdn), NK, and NKT cells in SSc patients. Similarly, Fox et al. [186] analyzed the PBMC of SSc patients by multi-parameter flow cytometry and found an increased percentage of CD4^+^T but a decreased percentage of CD8^+^T cells. In addition, the CD28 negative population expanded, and CD319^+^ T cells were strikingly expanded in the CD4^+^subset. It is interesting that a higher proportion of CD319^+^ cells producing a higher amount of IL-17 cytokine was observed. These CD4^+^CD319^+^(SLAM-F7^+^) cells are cytotoxic and are the dominant T cell population in perivascular lymphocytic infiltrate in SSc skin actively secreting IL-17 cytokine family. Moon et al. [187] unveiled that metformin, a medication used to treat diabetes, has effective immunomodulatory effects to inhibit IL-17 and Th17-related cytokines, IL-1β, IL-6, and TNF-α, via suppression on the phosphorylation of mTOR-STAT3 signaling.

Transcriptome analysis of immune cells from SSc patients by Paleja et al. [188] revealed that the activated CD4 phenotype was accompanied by an increased expression of inhibitory molecules while the reminiscent phenotype exhibited functional adaptation. The exhausted T cells were found in response to chronic stimulation. In conclusion, the systemic immunosome study highlights the potential pathological role of inflammation and chronic stimulation in mediating functional adaptation of immune-related cells and accelerated inflamm-aging in patients with SSc.

The abnormal immunopathological and immunobiological bases of accelerated inflamma-aging in patients with SSc are summarized in Figure 6.

## 6. Conclusions

Systemic sclerosis is a rare, idiopathic, and intricate systemic autoimmune disease. It is characterized by a triad of immune dysregulation, vasculopathy, and chronic inflammation that ends in tissue fibrosis. In addition, cancer incidence in SSc patients increased via chromosome instability and abnormal DNA damage responses. These features mimic inflamm-aging in the elderly. Although genetic predispositions in certain HLA and non-HLA loci are found, the concordance rate of SSc in monozygotic twin is quite low (4.2%) compared to the other autoimmune diseases. Accordingly, environmental and stochastic factors with subsequent aberrant epigenetic regulation would play crucial roles in SSc pathogenesis. The overproduction of oxidative/nitrodative radicals in causing DNA damage and abnormal DNA damage responses may further activate innate and adaptive immune responses in SSc patients. For compensating immune-inflammatory functions, an inflamm-aging state is induced for “immune adaptation.” It is rather important to clarify the exact signaling pathways resulting in accelerated inflamm-aging. The target therapy for these abnormal signaling pathways will become a useful therapeutic strategy for treating SSc patients in addition to the non-specific immunosuppressants.

## Figures and Tables

**Figure 1 cells-10-03402-f001:**
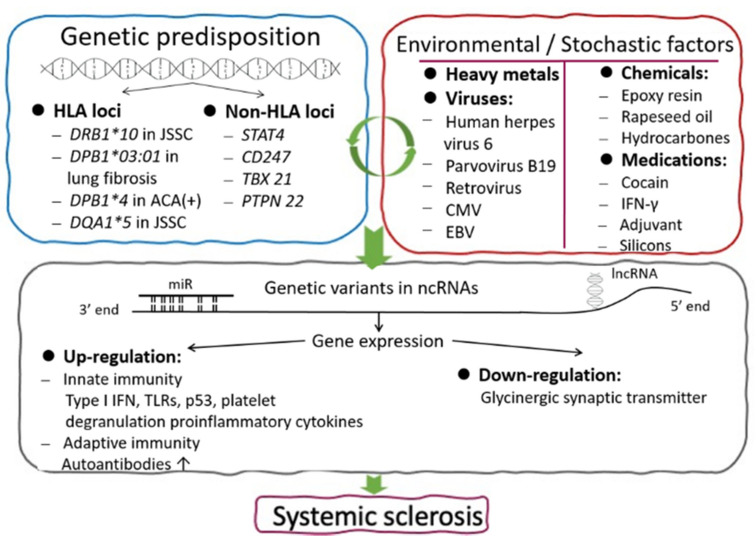
The intricate interactions among genetic predisposition, environmental/stochastic factors, and the epigenetic variants in the etiopathogenesis of patients with systemic sclerosis. Both HLA and non-HLA genetic loci are involved in the development of SSc. Many environmental and stochastic factors may derange epigenetic regulation on immune responses resulting in inflammatory cytokines and autoantibodies production in SSc patients. ncRNA: non-coding RNA; miR: microRNA with 20–24 nucleotide bases (nt) length; lncRNA: long non-coding RNA with > 200nt length.

**Figure 2 cells-10-03402-f002:**
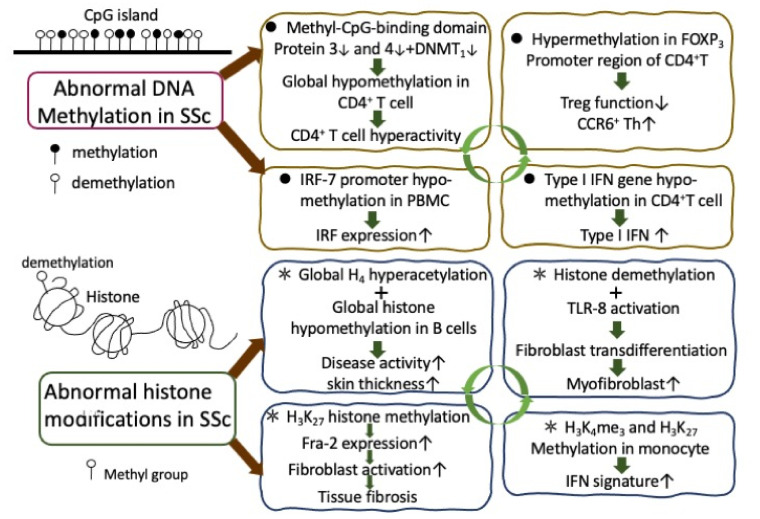
A number of abnormal epigenetic regulations on DNA methylation of the CpG islands in gene promoter regions (upper panel) and histone modifications (lower panel are found in patients with SSc). These deranged epigenetic regulations may result in chronic inflammation, immune dysfunction, and finally tissue fibrosis. IRF: interferon regulatory factor; IFN: interferon; TLR: Toll-like receptor; Treg: regulatory T cell; H: histone; K: lysine residue.

**Figure 3 cells-10-03402-f003:**
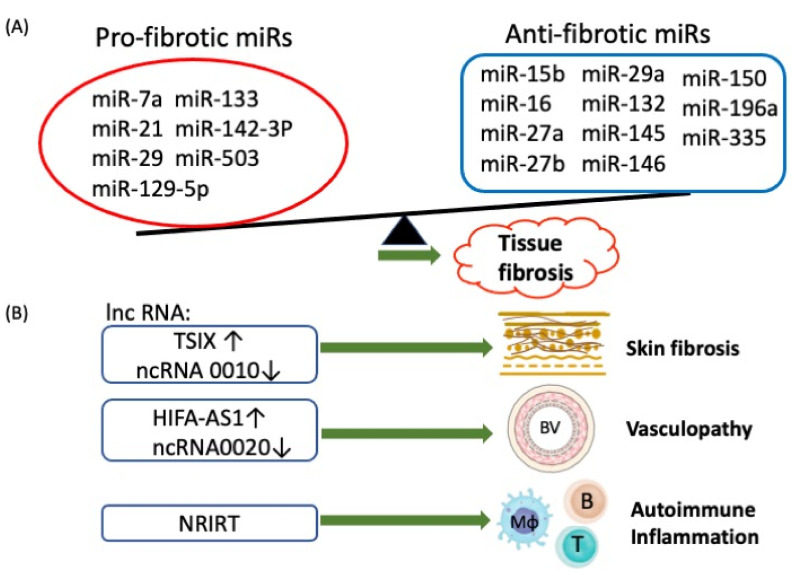
The abnormal epigenetic regulation by non-coding RNAs including microRNAs (miRs) and long non-coding RNAs (lncRNAs) in inducing pathological changes in patients with SSc. (**A**) The imbalance between profibrotic and anti-fibrotic microRNAs (miRs) expression shifts the effects toward tissue fibrosis. (**B**) The effects of individual lncRNA on the occurrence of pathological lesions in the patients with SSc; BV: blood vessel; Mφ: macrophage.

**Figure 4 cells-10-03402-f004:**
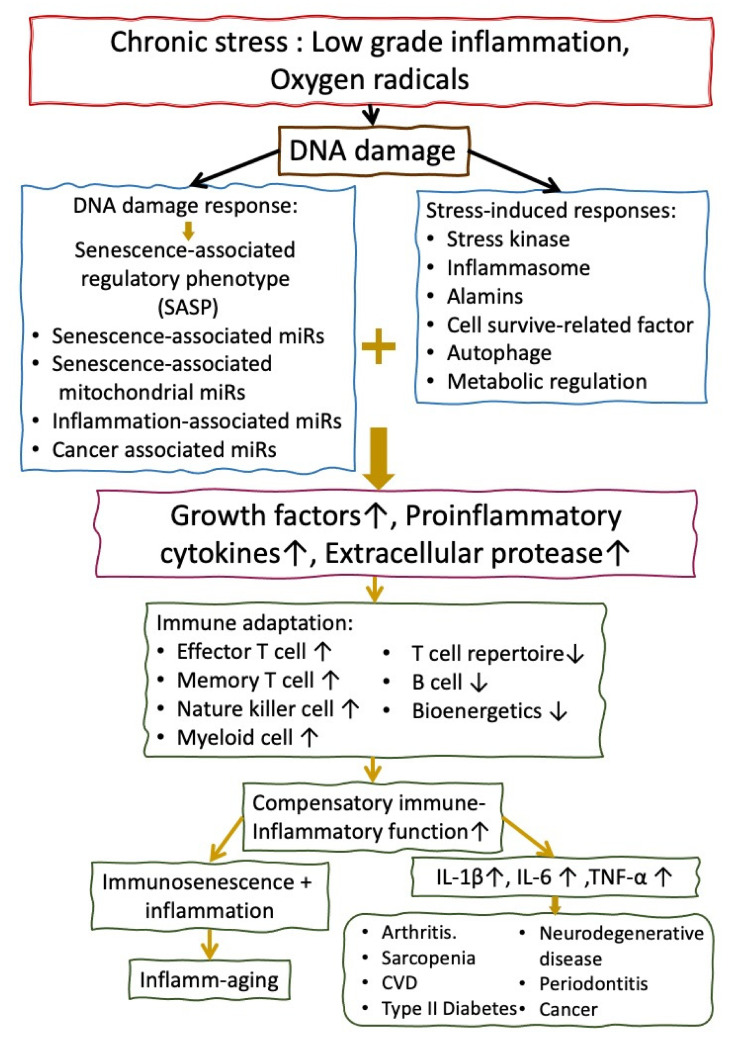
The chronic stresses induced by low-grade inflammation and increased oxygen radicals as upstream etiological factors in mediating abnormal DNA damage/DNA damage responses, immune adaptation manifested by compensatory immune-inflammatory function, and inflamm-aging in patients with SSc. Accelerated inflamm-aging in SSc may increase inflammation-associated diseases such as arthritis, sacropenia, cardiovascular diseases, neurodegenerative diseases, periodontitis, type II diabetes, and cancers via immune adaptation.

**Figure 5 cells-10-03402-f005:**
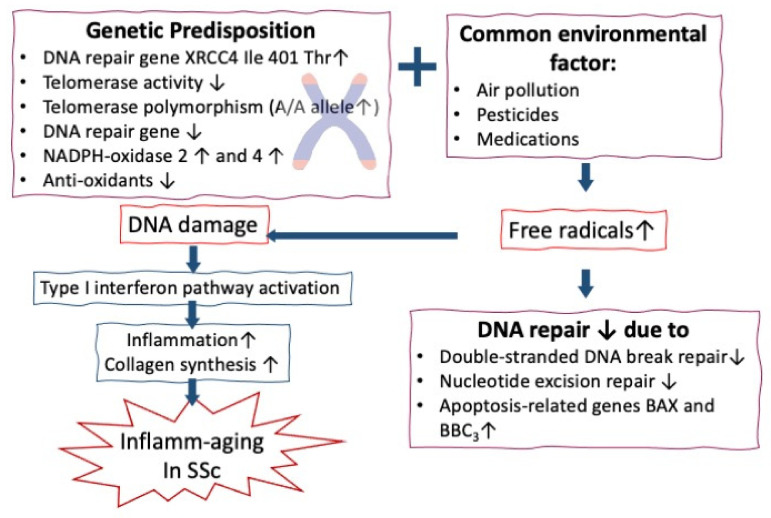
The genetic defects containing the DNA repair gene, telomerase activity, redox genes, and anti-oxidant capacity superimposed by common environmental factors are responsible for increased DNA damage by increased free oxygen radical production in SSc patients. Increased DNA damage then activates type 1 interferon pathway to induce inflammation and collage synthesis mimicking inflamm-aging in patients with SSc. On the other hand, the free radicals may also further impede DNA repair in SSc patients.

**Figure 6 cells-10-03402-f006:**
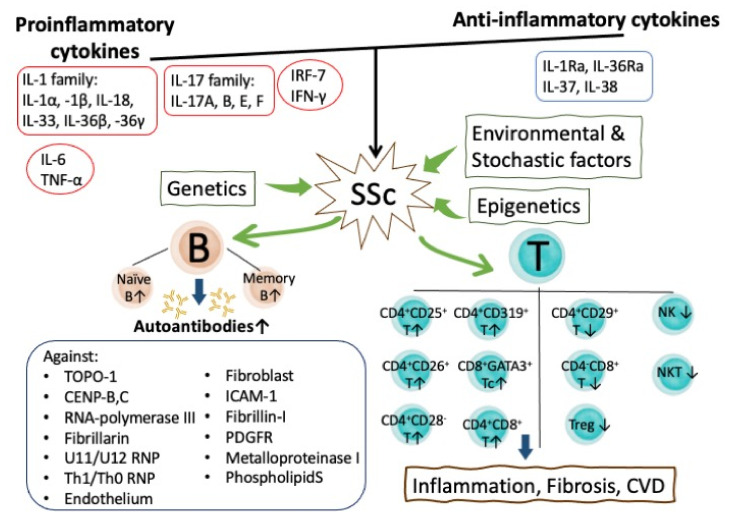
The roles of imbalanced proinflammatory/anti-inflammatory cytokines and deranged ontogenetic/immunobiological development of B and T lymphocytes in the induction of inflamm-aging of patients with SSc. The autoantibodies and abnormal T cell responsiveness are the two essential pathological factors for eliciting tissue inflammation, fibrosis, and cardiovascular complications in these patients. PDGF: platelet-derived growth factor; Tc: cytotoxic T cell; Treg: regulatory T cell; NK: natural killer cells; NKT: natural killer T cell.

## Data Availability

Not applicable.

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
