# Peer review of "Molecular Basis of Accelerated Aging with Immune Dysfunction-Mediated Inflammation (Inflamm-Aging) in Patients with Systemic Sclerosis"

_cells, 2021, doi:10.3390/cells10123402_

Round 1

Reviewer 1 Report

In the manuscript, Shen CH et al reviewed recent advances on the immunological basis of inflamm-aging in patients with SSc. The paper is clearly written, very interesting to read and timely.

Minor points:

- Figure 1: Please detail the following terms: miR, ncRNAs, IncRNA

- Line 313: typo: IFN-

Author Response

Answer Sheet

   Many thanks to the Editor and Reviewers to give us the opportunity to revise our review article (cells: 1437390) submitted for the special issue on “Inflammaging: the immunology of aging”. We took each comments, suggestions, and queries seriously and tried our best to answer them point-to-point. We used the “tract changes” function to revise the manuscript. In addition, the revised portions are underlined for easy recognition by the Reviewers. We cordially hope the revised version will be more acceptable by this world important journal in the field of cell research.

Answer to Reviewer (1):

Comment: In the manuscript, Shen CH et al. reviewed recent advances on the immunological basis of inflamm-aging in patients with SSc. The paper is clearly written, very interesting to read and timely.

Response: Thanks for the positive comment. We will work more in future.

Question (1): Minor point: Figure 1: Please detail the following terms: miR, ncRNAs, IncRNA

Answer: Thanks for the reasonable criticism. We have explained in detail in the legend of figure 1 in P.4 that;

     “ncRNA: non-coding RNA; miR: microRNA with 20-24 nucleotide bases (nt) length; lncRNA: long non-coding RNA with >200nt length”

Question (2): Minor point: Line 313: typo: IFN-

Answer: Thank you for pointing-out the typo in IFN-g and IFN-a. We have already corrected it in P.11-Section 4.3

Looking forward to hearing from you soon.

Yours truly,

 Chieh-Yu, Shen, MD

 Chia-Li Yu, MD, PhD

 Department of Internal Medicine

 National Taiwan University Hospital

 Taipei 1002, Taiwan

Reviewer 2 Report

The topic, Molecular Basis of Accelerated Aging with Immune Dysfunction-mediated Inflammation (Inflamm-aging) in Patients with Systemic Sclerosis is important and may interest the Cells’ readers.

This review helps to understand the molecular basis of accelerated aging in patients with scleroderma. The main content of the review is well-written and well-understandable. The Figures are well understood as well and complement those described in the study.

However, no description of the method of screening the studies and extraction of data is given (e.g. independent reviewers, management of eventual disagreements). No reference to the PRISMA guideline is given, some of its key points and the flow diagram are missing.

I have problems concerning figure 4

What is „Alamins” ?  mention of it is missing in the text.

There are spelling mistakes: e.g.  „inflammation-associated „  „Diabetes” on figure 4  

in line 292 „β” is  missing:  by transforming growth factor β (TGF-).  

Not much practical value for practicing physicians but helps and supports causes in gaining attention and in understanding the pathogenesis of SSc. 

Author Response

Answer Sheet

   Many thanks to the Editor and Reviewers to give us the opportunity to revise our review article (cells: 1437390) submitted for the special issue on “Inflammaging: the immunology of aging”. We took each comments, suggestions, and queries seriously and tried our best to answer them point-to-point. We used the “tract changes” function to revise the manuscript. In addition, the revised portions are underlined for easy recognition by the Reviewers. We cordially hope the revised version will be more acceptable by this world important journal in the field of cell research.

Answer to Reviewer (2):

Comment:  The topic, Molecular Basis of Accelerated Aging with Immune

         Dysfunction-mediated Inflammation (Inflamm-aging) in Patients with

         Systemic Sclerosis is important and may interest the Cells’ readers.

This review helps to understand the molecular basis of accelerated aging in patients with scleroderma. The main content of the review is well-written and well-understandable. The Figures are well understood as well and complement those described in the study.

However, no description of the method of screening the studies and extraction of data is given (e.g. independent reviewers, management of eventual disagreements). No reference to the PRISMA guideline is given, some of its key points and the flow diagram are missing.

Answer: Thanks for this important criticism. As a matter of fact, we do not really understand the meaning of this criticism. We have read the principle of the “PRISMA-P” and found it is preferred for reporting items for systemic review and meta-analysis protocol. The present review article only stated or summarized the results from the published literatures, but not conducted a meta-analytic study from these published data. Our review article is quite simple only for reviewing the published literatures and make a summary or conclusion for them. We need your advanced suggestion for accomplish this comment, if possible.

Qustion (1): I have problems concerning figure 4, What is “Alamins” ? mention of it is missing in the text.

Answer: We are extremely sorry for omitting the definition of the “alamins” in text. In the revised version, we have already added a statement of alamins in P.7 Section 4 for more clarity that;

          Alamins represent another emerging potential family of inflammation- associated early phase SASP regulators. These molecules are easily released from the chromatin and actively secreted at senescence or during cell necrosis. The chromatin-associated high mobility group box 1 protein (HMGB1) is the first found alamin acting as both both early phase SASP regulator and SASP factor [101].

Question (2): There are spelling mistakes: e.g. “inflammation-associated Diabetes” on figure 4 

Answer: We are extremely sorry for this spelling mistake in Fig.4. We have already corrected it and other spelling errors in text.

Question (3): In line 292 „β” is missing:  by transforming growth factor β (TGF-). 

Answer: Thanks again for pointing-out our spelling errors due to unknown reason. We have already corrected them in the text of the new version.

Question (4): Not much practical value for practicing physicians but helps and supports causes in gaining attention and in understanding the pathogenesis of SSc.

Answer: Thank you for this elegant comment.  

Looking forward to hearing from you soon.

Yours truly,

 Chieh-Yu, Shen, MD

 Chia-Li Yu, MD, PhD

 Department of Internal Medicine

 National Taiwan University Hospital

 Taipei 1002, Taiwan

Reviewer 3 Report

This an interesting review which describes extensively the so far accumulated evidence on inflammaging in Systemic Sclerosis.  This is a hot topic for this devastating disease and the authors should be complimented for their effort.  However I believe that the ms is too long  (for example section #4) and a reduction of at least 40 % would be appropriate. Also, I suggest that strong wording (i.e undoubtedly, obviously, etc0 should be avoided.  Finally, the contribution of vasculopathy may deserve some additional discussion    

Author Response

Answer Sheet

   Many thanks to the Editor and Reviewers to give us the opportunity to revise our review article (cells: 1437390) submitted for the special issue on “Inflammaging: the immunology of aging”. We took each comments, suggestions, and queries seriously and tried our best to answer them point-to-point. We used the “tract changes” function to revise the manuscript. In addition, the revised portions are underlined for easy recognition by the Reviewers. We cordially hope the revised version will be more acceptable by this world important journal in the field of cell research.

Answer to Reviewer (3):

Comment: This is an interesting review which describes extensively the so far accumulated evidence on inflammaging in Systemic Sclerosis. This is a hot topic for this devastating disease and the authors should be complimented for their effort. However I believe that the ms is too long  (for example section #4) and a reduction of at least 40 % would be appropriate. Also, I suggest that strong wording (i.e undoubtedly, obviously, etc0 should be avoided. Finally, the contribution of vasculopathy may deserve some additional discussion.

Response: Many thanks for these important suggestions and criticism. In the revised version, we have already deleted the inappropriate wording including “undoubtedly” and “obviously” in “Abstract” and text. In addition, we have tried our best to reduce the Section 4. However, we can only reduce around 30% statements of the Section 4 since some descriptions for the results, we consider, are important for the completion of literature reviewing. If the Reviewer still insists for more reduction and the English is required for editing we will do it again.

            We also discuss more on the contribution of vasculopathy in the “Introduction” Section (P. 1) in that;

              It is conceivable that inflammatory cytokines (IL-1b, IL-6 and   TNF-a) would enhance oxidative stresses to cause endothelial cell dysfunction and mitochondrial functional impairment. This early vascular aging may further accelerate inflamm-aging in patients with SSC.

          The above statements in conjunction with the statements in the “Introduction” as shown below can become a discussion of the contribution of vasculopathy in accelerated inflamm-aging in patients with SSc.

“In clinical practice, the endothelial dysfunction manifested as Raynaud’s phenomenon is the first clinical presentation in the patients due to damage in the finger capillary lumen [6, 7]. Furthermore, the vascular injury may induce fibroblast activation and tissue fibrosis leading to irreversible scarring and organ failure [8-10].”

Looking forward to hearing from you soon.

Yours truly,

 Chieh-Yu, Shen, MD

 Chia-Li Yu, MD, PhD

 Department of Internal Medicine

 National Taiwan University Hospital

 Taipei 1002, Taiwan

Round 2

Reviewer 2 Report

Thank you for answering the questions and thank you for all corrections. This  review helps to understand the molecular basis of accelerated aging in patients with scleroderma. The content of the review is well written and understandable. I recommend the publication of the manuscript in present form.